# Application of Texture and Fractal Dimension Analysis to Evaluate Subgingival Cement Surfaces in Terms of Biocompatibility

**DOI:** 10.3390/ma14195857

**Published:** 2021-10-07

**Authors:** Katarzyna Skośkiewicz-Malinowska, Martyna Mysior, Agnieszka Rusak, Piotr Kuropka, Marcin Kozakiewicz, Kamil Jurczyszyn

**Affiliations:** 1Department of Conservative Dentistry with Endodontics, Wroclaw Medical University, 50-425 Wroclaw, Poland; Katarzyna.skoskiewicz-malinowska@umed.wroc.pl; 2SCTT Academic Dental Polyclinic, 50-425 Wroclaw, Poland; 3Division of Histology and Embryology, Department of Human Morphology and Embryology, Wroclaw Medical University, 50-368 Wroclaw, Poland; agnieszka.rusak@umed.wroc.pl; 4Division of Histology and Embryology, Veterinary Medicine, Wroclaw University of Environmental and Life Sciences, 50-375 Wroclaw, Poland; piotr.kuropka@upwr.edu.pl; 5Department of Maxillofacial Surgery, Medical University of Lodz, 90-647 Lodz, Poland; marcin.kozakiewicz@umed.lodz.pl; 6Department of Dental Surgery, Wroclaw Medical University, 50-425 Wroclaw, Poland; kamil.jurczyszyn@umed.wroc.pl

**Keywords:** fractal dimension analysis, texture analysis, dental cements, subgingival restoration, fibroblasts adhesion, cytotoxicity

## Abstract

Biocompatibility is defined as “the ability of a biomaterial, prosthesis, or medical device to perform with an appropriate host response in a specific application”. Biocompatibility is especially important for restorative dentists as they use materials that remain in close contact with living tissues for a long time. The research material involves six types of cement used frequently in the subgingival region: Ketac Fil Plus (3M ESPE, Germany), Riva Self Cure (SDI, Australia) (Glass Ionomer Cements), Breeze (Pentron Clinical, USA) (Resin-based Cement), Adhesor Carbofine (Pentron, Czech Republic), Harvard Polycarboxylat Cement (Harvard Dental, Great Britain) (Zinc polycarboxylate types of cement) and Agatos S (Chema-Elektromet, Poland) (Zinc Phosphate Cement). Texture and fractal dimension analysis was applied. An evaluation of cytotoxicity and cell adhesion was carried out. The fractal dimension of Breeze (Pentron Clinical, USA) differed in each of the tested types of cement. Adhesor Carbofine (Pentron, Czech Republic) cytotoxicity was rated 4 on a 0–4 scale. The Ketac Fil Plus (3M ESPE, Germany) and Riva Self Cure (SDI, Australia) cements showed the most favorable conditions for the adhesion of fibroblasts, despite statistically significant differences in the fractal dimension of their surfaces.

## 1. Introduction

Biocompatibility is defined as “the ability of a biomaterial, prosthesis, or medical device to perform with an appropriate host response in a specific application” [1]. Because no material can be proven to be 100% biologically safe, technical tests and data are required to determine when the benefits outweigh the risks [2]. There is a sequence of studies assessing the safety of new materials. One of these is in vitro research, followed by investigations conducted on animals and, finally, clinical studies. This makes it possible to evaluate the biocompatibility of new materials and eliminate those with more significant cytotoxic potential [3].

Biocompatibility is especially important for restorative dentists as they use materials that remain in close contact with living tissues for a long time [2]. The increasing variety of substances used in dentistry is associated with the technological complexity of more recent materials. This often leads to an increased awareness of biological effects and narrows down the possibilities of using those materials. Unfortunately, adverse biological reactions to materials used in dentistry are often unnoticed by patients or dentists [4]. It is especially relevant when they occur in the subgingival area because of their potentially negative impact on biofilm accumulation, direct irritation of the gingiva, and damage to the biological width [5]. Consequently, it is a significant challenge for the restorative material to incorporate the hard tissue of the tooth and the soft tissue of the gingiva [6].

Biocompatibility of restorative materials, marginal seal, marginal fit, emergence profile, and elemental release are the main factors that have an impact on the health of the marginal gingiva [7]. Gingival fibroblasts are the major population of the gingival connective tissue. They are important for periodontal health as they are responsible for the synthesis and degradation of the extracellular matrix, which plays a critical role in tooth anchorage and wound healing [8]. Subgingival restorations may result in an increased accumulation of plaque, gingival inflammation, periodontal destruction, increased pocket depth, loss of attachment, and gingival recession [9].

Among the materials placed close to the gingiva, or even subgingivally, are glass ionomer cements (GIC) [10,11,12,13]. Glass ionomers were first introduced in 1978 as aluminosilicate poly acrylic cement (ASPA), which can be specifically applied in atraumatic treatment [6]. The most valuable features of glass ionomer cement include fluoride release [14], the coefficient of thermal expansion compatible with tooth tissues [15], the possibility to use them in cavities without the need for bonding agents [16], and the formation of direct chemical adhesion to the tooth structure [17]. On the other hand, the lack of sufficient strength and toughness seem to constitute the most important disadvantages of glass ionomers [18]. The types of cement that are based on glass ionomer materials are extremely popular definitive luting agents. The chemistry of the setting reaction is complex and takes several months to reach completion [19].

Due to their mechanical and aesthetic properties, resin-based materials are widely used in modern dentistry as direct filling materials. They are also used in the proximity of the gingival area, often to restore noncarious cervical lesions [20]. Resin-based materials are used also as luting cements, often in subgingival preparations with challenging field isolation. The greatest advantages of resin luting materials include improved mechanical properties, lower solubility, and reinforcement of all-ceramic restorations in comparison with the traditional luting cements [21]. Self-adhesive resin cements are based on filled polymers created to adhere to the tooth structure without the need to use a separate adhesive or etchant. Basic components of these materials include an organic matrix with phosphoric acid methacrylates or acidic monomers [22].

Zinc phosphate has been routinely used for 100 years, and it is the standard material to which other types of cement are often compared [23]. It is the classic AB cement whose powder contains 90% zinc oxide (ZnO), and the liquid is made of 67% buffered phosphoric acid. The pH of zinc phosphate 1 h after delivery amounts to less than 4 and reaches a neutral value after 48 h [19]. Due to its early strength, low cost, acceptable physical properties, and easy clinical use, zinc phosphate is used for luting fixed partial dentures including metal–ceramic restorations and cast post cores [24].

Zinc polycarboxylate cement was introduced as the first luting cement that would adhere to the tooth structure [19]. Zinc polycarboxylate cement is formed by heat-treated zinc oxide and aqueous polyacrylic acid [25]. Chemical adhesion is best for the enamel, and it is caused by the interaction of free carboxylic groups and calcium [26]. The pH of zinc polycarboxylate cement is more acidic than that of zinc phosphate at first, but it rises faster to a neutral value.

When considering biocompatibility studies, the most widely used tests examine cell viability. These tests involve the examination of the mitochondrial activity or the occurrence of necrosis or apoptosis of a cell exposed to various materials [27]. The cells that can be easily tested and are commonly used for cytotoxicity tests in dentistry are fibroblasts [28]. Normal fibroblast function is necessary to obtain periodontal tissue function.

There are certain limitations to the study and mathematical analysis of actual structures. These stem from the complicated and irregular construction of real objects visible at various scale levels. One of the methods used to analyze complex things is texture analysis (TA) and fractal dimension analysis (FDA).

A pixel is the smallest element of an image representing a specific color that is presented in digital photographs. Texture can be defined as an exemplary structure of an image that is created by pixels. A group of repetitive graphical attributes, such as brightness, smoothness, entropy, coarseness, linearity, or regularity, can characterize texture. Texture analysis provides quantitative, accurate, and sensitive detection of subtle changes in tested structures. TA comprises a series of mathematical techniques used to quantify and evaluate spatial variations in pixel grayscale intensities within a digital image [29]. Texture analysis is an essential tool used in various cases as object recognition, surface defect detection, pattern recognition, and medical image analysis. The classification of texture analysis methods highlights four categories: statistical methods, structural, model-based, and transform-based strategies [30]. TA is frequently used in dentistry for the analysis of computed tomography, magnetic resonance, or X-ray images [31,32,33].

A fractal is a geometrical object introduced by Mandelbrot that deals with self-similar forms compared to typical geometries taught in Cartesian and Euclidean mathematics [34]. While objects defined in Euclidean geometry have an integer topological dimension, fractals are characterized by an index of structural complexity called fractional dimension (FD). The difference between these two types and their FD is presented in Figure 1. Fractal dimension analysis (FDA) has provided a mathematical formalism for describing complex spatial and dynamical structures [35]. Self-similarity is commonly found in complex natural objects. Fractal dimension analysis based on image processing is broadly used in many areas of knowledge, including medicine, dentistry, technology, and materials science [34,36]. In materials, the fractal dimension is the most relevant parameter of surface topography [37]. It can provide information on subtle structural changes, or even mechanisms leading to its formation. The topography of a surface can be examined using such imaging methods as light microscopy, atomic microscopy (AFM), scanning tunneling microscopy (STM), and electron microscopy (SEM) [33]. FDA is broadly used in surface testing of such dental materials as lithium disilicate-based crowns [38], zirconia dental implants [39], or dental restorative composite [40].

This study aims to apply texture and fractal dimension analysis to evaluate subgingival cement surfaces in terms of biocompatibility.

## 2. Materials and Methods

### 2.1. Procedure for Preparing the Cement Samples

The research material was prepared using six cements frequently applied in the subgingival region: Glass Ionomer Cements: Ketac Fil Plus (3M ESPE, Seefeld, Germany), Riva Self Cure (SDI, Australia), Resin-based Cement: Breeze (Pentron Clinical, Wallingford, CT, USA), Zinc polycarboxylate cements: Adhesor Carbofine (Pentron, Chodov, Czech Republic), Harvard Polycarboxylat Cement (Harvard Dental, Great Britain), Zinc Phosphate Cement: Agatos S (Chema-Elektromet, Poland). The cements were prepared in the form of 0.7 cm × 0.7 cm cubes based on a matrix made of the plastic presented in Figure 2. The materials were prepared in line with the manufacturers’ recommendations and then sterilized with ethylene oxide gas. The 2.5 h EOG sterilization cycle was performed at 55 °C, and then the research material was subjected to a 12 h degassing process.

### 2.2. Biological Evaluation

#### 2.2.1. Cell Line

The Balb/3T3 cell lines were cultured using the DMEM medium (Lonza, Basel, Switzerland) with 10% fetal bovine serum (FBS) and 1% L-glutamine with a penicillin and streptomycin solution (Sigma-Aldrich^®^, St. Louis, MO, USA). Normal human dermal fibroblasts NHDF (Lonza) were cultured in the FGM^TM^ Fibroblast Growth Medium BulletKit^TM^ (Lonza). Cell cultures were conducted at 37 °C, 5% of CO_2_ with constant air humidity in the HERA cell CO2 Incubator 150i (Thermo Scientific, Waltham, MA, USA).

#### 2.2.2. Determination of Cytotoxicity

Determination of cytotoxicity was conducted using normal Balb/3T3 murine fibroblasts (American Type Culture Collection ATCC^®^, Old Town Manassas, VA, USA), which is one of the in vitro models used in the biological evaluation of medical devices [41,42,43,44]. The cells were seeded on 6-well plates in the amount of 1.0 × 10^5^ cells per well. After 24 h of culture, the medium was changed, the material was applied to each well, and the culture was conducted for 24 h. Subsequently, an evaluation using the Olympus CKX53 contrast-phase inverted microscope (Olympus, Tokyo, Japan) was conducted. Morphology of cells was assessed under the specimens, around the materials, and in the whole well. The cytotoxic effect was determined according to a four-grading scale where changes in the culture over grade 2 (mild grade) are considered to be the cytotoxic effect [41]. A cell culture without contact with the evaluated materials constituted the control in the study.

#### 2.2.3. Cell Adhesion

The NHDF cells were carried out on materials in 6-well plates in the amount of 6 × 10^4^ cells and cultured for 48 h and five days. During this time, the cell cultures were gently rocked to evaluate the actual adhesion of cells to the surface of the material. Subsequently, the cells were stained with a mixture of DAPI, 0.1µg/mL (Thermo Fisher, Waltham, MA USA), and propidium iodide, 0.5 mg/mL (Roche, Mannheim, Germany), and visualized with Eclipse80i fluorescence microscope (Nikon Corporation, Tokyo, Japan). Cells without contact with the evaluated material constituted the control in the study.

### 2.3. Taking Images

All photographs were taken using the Techrebal K10E stereoscopic microscope (Techrebal, Wilczyce, Poland). The eyepiece was replaced by the ZWO ASI178mm monochrome digital camera (ZWO CO., LTD., Suzhou, China). All photographs were taken using an 18× magnification and 3096 × 2080 resolution. The time of exposure was set to achieve histogram filling at the level of 90%, depending on the cement surface. The gain parameter (the sensitivity of CMOS matrix) was the same during all procedures, and it was set to 10 to reduce the noise. A camera in a 14-bit mode was used to achieve the widest dynamic range of photographs. The images were saved as 16-bit TIFF (Tagged Image File Format) files. Two graphical operations were applied to normalize all images for further analysis: auto-levels and high-pass filter to decrease the effect of non-homogeneous illumination of the examined samples (Figure 3). Subsequently, the images were saved as 8-bit grayscale bitmaps. All graphical operations were performed using GIMP, version 2.10.24 (GNU Image Manipulation Program—www.gimp.org, free and open source license, accessed 1 October 2021).

### 2.4. Fractal Dimension Analysis

All fractal analyses were performed in ImageJ, version 1.53e (Image Processing and Analysis in Java—Wayne Rasband and contributors, National Institutes of Health, USA, public domain license, https://imagej.nih.gov/ij/ accessed 1 October 2021), and plugin FracLac, version 2.5 (Charles Sturt University, Australia, public domain license, accessed 1 October 2021).

In our study, we decided to use a modified algorithm of the counting box method, which makes it possible to analyze monochromatic images, such as 8- or 16-bit images. In the case of grayscale images, we applied the intensity difference algorithm to calculate fractal dimension. The analyzed image is divided into boxes as in the counting box method. The difference between the maximum pixel intensity and the minimum pixel intensity is calculated in each box (δI_i,j,ε_, where i, j—the location of the analyzed box in the ε scale)
δI_i,j,ε_ = maximum pixel intensity_i,j,ε_ − minimum pixel intensity_i,j,ε_(1)

In the next step, 1 is added to the intensity difference to prevent its value from becoming a 0
I_i,j,ε_ = δI_i,j,ε_ + 1(2)

Finally, the fractal dimension of the intensity difference is described using the following formula
(3)$$\mathrm{FD}=\underset{\varepsilon \to 0}{\lim}\frac{\ln\left(I\varepsilon \right)}{\ln\left(\frac{1}{\varepsilon }\right)}$$
where FD—final fractal dimension of intensity, Iε = Σ[1δIi,j,ε + 1], ε—scale of box.

All operations are shown in Figure 4.

### 2.5. Texture Analysis

The region of interests (ROIs) were normalized (μ ± 3σ) to share the same average (μ) and standard deviation (σ) of optical density within the ROI for texture analysis. Selected image texture features (entropy and difference entropy from the co-occurrence matrix, and long-run emphasis moment from the run-length matrix) in ROIs were calculated
(4)$$\mathrm{Entropy}=-\sum_{i=1}^{\mathrm{Ng}}\sum_{j=1}^{\mathrm{Ng}}p\left(i,j\right)\log(p\left(i,j\right))$$
(5)$$\mathrm{DifEntr}=-\sum_{i=1}^{\mathrm{Ng}}p_{x-y}\left(i\right)\log\left(p_{x-y}\left(i\right)\right)$$
where Σ is the sum, Ng is the number of grey levels in the microphotograph, i and j stand for a grey level of pixels 5-pixel distant one from another, p is the probability, log is the common logarithm.
(6)$$\mathrm{LngREmph}=\frac{\sum_{i=1}^{\mathrm{Ng}}\sum_{k=1}^{\mathrm{Nr}}k^{2}p\left(i,k\right)}{\sum_{i=1}^{\mathrm{Ng}}\sum_{k=1}^{\mathrm{Nr}}p\left(i,k\right)}$$
where Σ is the sum, Nr is the number of series of pixels with grey level i and length k, Ng is the number of grey levels for the image, Nr is the number of pixels in series, p is probability [45,46]. These three equations were subsequently used for the texture index construction [47]. Finally, the texture index (TI) and bone index (BI), which represent the ratio of the measure of the diversity of the structure observed in the microphotograph to the measure of the presence of uniform longitudinal structures, was calculated
(7)$$\mathrm{Texture}\;\mathrm{index}=\frac{\mathrm{Entropy}}{\mathrm{LngREmph}}=\frac{(-\sum_{i=1}^{\mathrm{Ng}}\sum_{j=1}^{\mathrm{Ng}}p\left(i,j\right)\log\left(p\left(i,j\right)\right))\sum_{i=1}^{\mathrm{Ng}}\sum_{k=1}^{\mathrm{Nr}}p\left(i,k\right)}{\sum_{i=1}^{\mathrm{Ng}}\sum_{k=1}^{\mathrm{Nr}}k^{2}p\left(i,k\right)}$$
(8)$$\mathrm{Bone}\;\mathrm{Index}=\frac{\mathrm{DifEntr}}{\mathrm{LngREmph}}=\frac{(-\sum_{i=1}^{\mathrm{Ng}}p_{x-y}\left(i\right)\log\left(p_{x-y}\left(i\right)\right))\sum_{i=1}^{\mathrm{Ng}}\sum_{k=1}^{\mathrm{Nr}}p\left(i,k\right)}{\sum_{i=1}^{\mathrm{Ng}}\sum_{k=1}^{\mathrm{Nr}}k^{2}p\left(i,k\right)}$$

The index defined in this way [48] was taken as a measure of cement surface structure.

### 2.6. Statistical Analysis

Statistica version 13.3 (StatSoft, Cracow, Poland) was used to perform all statistical tests. A value of 0.05 was deemed to be statistically significant. The Shapiro–Wilk test was used to confirm the normality of distribution. Due to a normal distribution, parametric tests were performed. The analysis of variance (ANOVA) and the least significant difference post hoc were applied to reveal the fractal dimension differences between the examined surfaces of cements. A correlation matrix was used to estimate the correlation between FD’s measured surface and the adhesion of fibroblasts. Qualitative properties of cells were converted to quantitative parameters. For amounts of cells: lack of cells—0, few cells—1, cells on surface—2, culture of cells—3; for cytotoxicity: none—0, moderate—1, severe—2. Correlation coefficient were calculated two times between FD, TI, BI and amounts of cells (after 48 h and 5 days). The value of this coefficient was lower after 5 days than after 48 h

## 3. Results

### 3.1. Fractal Dimension

Table 1 presents a summary of the mean value of the fractal dimension for individual cements with the results of cell adhesion and cytotoxicity. Our study revealed an average correlation coefficient (r = 0.47) between the fractal dimension and the number of fibroblasts after 48 h. After 5 days of fibroblast incubation, the correlation coefficient dropped to 0.31. Figure 5 shows examples of light microscope images of the surfaces of the investigated cements subjected to a subsequent analysis of textures and the fractal dimension.

Table 2 presents the results of the post hoc ANOVA test (least significant difference) for the differences in the fractal dimensions of particular cement surfaces. The fractal dimension of Adhesor Carbofine (Pentron, Czech Republic) was statistically different than all cements except for Harvard Polycarboxylat Cement (Harvard Dental, Great Britain) and Ketac Fil Plus (3M ESPE, Germany). The structure of the surface of Agatos S (Chema-Elektromet, Poland) did not show any differences apart from in relation to Harvard Polycarboxylat Cement (Harvard Dental, Great Britain). The fractal dimension of Breeze (Pentron Clinical, USA) was different from every tested cement. The fractal dimension of the surface of Harvard Polycarboxylat Cement (Harvard Dental, Great Britain) showed statistical differences only in relation to the earlier-mentioned Breeze (Pentron Clinical, USA). The structure of the surface of Ketac Fil Plus (3M ESPE, Germany) showed differences in the fractal dimension in relation to Agatos S (Chema-Elektromet, Poland), Breeze (Pentron Clinical, USA) and Riva Self Cure (SDI, Australia). There were no statistical differences between Riva Self Cure (SDI, Australia), Agatos S (Chema-Elektromet, Poland), and Harvard Polycarboxylat Cement (Harvard Dental, Great Britain).

### 3.2. Biological Evaluation

Agatos S (Chema-Elektromet, Poland), Harvard Polycarboxylat Cement (Harvard Dental, Great Britain) and Breeze (Pentron Clinical, USA) show moderate cytotoxicity (grade 3 on a 0–4 scale). 

Altered cells morphology was observed under and near the sample, such as lysis, rounded, vacuolated cells. In the rest well cell, morphology was normal, comparable to that of cells in the control culture that had no contact with the test materials.

Riva Self Cure (SDI, Australia) and Ketac Fil Plus (3M ESPE, Germany) showed a 0 degree of cytotoxicity and did not change the cell culture.

Cytoxicity of Adhesor Carbofine (Pentron, Czech Republic) was rated 4 on a 0–4 scale. After the contact with this material, the cell culture showed features of a disrupted culture.

According to PN-EN ISO 10993-5: 2009 Biological evaluation of medical devices—Part 5: Tests for In vitro cytotoxicity, medical devices with a cytotoxicity grade above 2 are considered to have a cytotoxic effect [41]. Morphology of Balb/3T3 after direct contact with the materials is presented in Figure 6 and Figure 7 and Table 3.

The investigated materials showed different adhesive properties. Riva Self Cure (SDI, Australia) and Ketac Fil Plus (3M ESPE, Germany) had the most favorable properties, promoting adhesion of the NHDF cells to the surface. Cells were also observed on the surface of Agatos S (Chema-Elektromet, Poland) but in a smaller amount. No cells were observed on the surface of Breeze (Pentron Clinical, USA) and Adhesor Carbofine (Pentron, Czech Republic). The conducted research also revealed that there is a relationship between the cytotoxicity of materials and their adhesive properties. In the case of Riva Self Cure (SDI, Australia) and Ketac Fil Plus (3M ESPE, Germany) that showed no cytotoxic effect, Riva Self Cure (SDI, Australia) appeared to have a little better adhesive properties than Ketac Fil Plus (3M ESPE, Germany) as cell clusters were observed after 5 days of cultivation, which may indicate that the material promotes coating by the cell layer. This could suggest that the NHDF cells proliferate on this surface (Figure 8).

### 3.3. Texture Analysis

Microphotographs of cement surfaces revealed significant differences among tested materials (Table 4 and Figure 9). The highest values of a texture index and Bone Index were noted in the case of Ketac Fil Plus (3M ESPE, Germany), Adhesor Carbofine (Pentron, Czech Republic), and Riva Self Cure (SDI, Australia). The lowest values of these features were noted in the case of Breeze (Pentron Clinical, USA) and Agatos S (Chema-Elektromet, Poland).

After 48 h of incubation, a weak correlation between the developed surface structures (higher TI and BI values) and the number of adherent cells to this surface can be observed (TI: R-squared = 19%; correlation coefficient = −0.44; *p* < 0.05. BI: R-squared = 15%; correlation coefficient = −0.38; *p* < 0.05). Simple regression equations: TI = sqrt(3.55419 + 0.24121 × Number of Cells^2^), BI = exp(−0.300535 + 0.0314068 × Number of Cells^2^), respectively.

The R-Squared statistic indicates that the model as fitted explains 8.5% of the variability in TI (*p* < 0.05) at the fifth day of incubation. The correlation coefficient equals −0.29, indicating a relatively weak relationship between the variables. The more developed the surface, the lower the cytotoxicity (in other words, more living cells adhere to the more developed surface: TI = 1.87042 + 0.0389976 × Number of Cells^2^). A similar relationship, and a weak one, was found for descriptions of surface structure using BI: R-Squared = 7%; correlation coefficient = −0.27; BI = 1/(1.37535 − 0.0278543 × Number of Cells^2^; *p* < 0.05).

## 4. Discussion

In this study, glass ionomer materials (Ketac Fil Plus (3M ESPE, Germany), Riva Self Cure (SDI, Australia)) showed no changes in fibroblast growth. The results are consistent with the majority of publications where glass ionomer cement showed low cytotoxicity [28,49,50]. Some research shows that the cytotoxic effects of GI cement are time-dependent. Mallineni et al. noticed that freshly-prepared GI materials are mildly cytotoxic but their cytotoxicity decreases over time. The reason for this phenomenon is fluoride release, which has a therapeutic value but causes cytotoxicity [4]. A study conducted by Lang et al. confirmed that viability and proliferation of fibroblast cells changed with time and showed no differences when compared to the control group by the 21st day of the study [51]. A negative biological influence of GI cement was demonstrated in a study by Milhem et al. It revealed that GI cement showed greater cytotoxicity than other materials, including composites [52].

The results of this study indicate that the highest cytotoxicity is shown by Zinc Polycarboxylate Cements, Adhesor Carbofine (Pentron, Czech Republic) and Harvard Polycarboxylat Cement (Harvard Dental, Great Britain). The cytotoxic effect of this group of cements has been confirmed by studies. A study conducted by Schmid-Schwap et al. suggests that the release of zinc ions and acidity may be the cause of cytotoxicity [53].

In this study, the resin-based material and zinc phosphate cements Breeze (Pentron Clinical, USA) and Agatos S (Chema-Elektromet, Poland) are characterized by moderate cytotoxicity. Some studies have shown that the resin-based materials are more cytotoxic than others. Monomers such as TEGDMA, HEMA, BisGMA, and UDMA reduce cell viability and generate a breakdown of the mitochondrial membrane in fibroblasts [54]. A study conducted by De Souza Costa et al. showed that light-cured resins are less cytotoxic than chemically-cured systems. The results depend on the curing efficiency of the polymerization lamp and the type of the resin system [3].

Fractal analysis was used by Salareno et al. to investigate the effect of air polishing on the surface of composites. They concluded that the use of glycine in air polishing generates the least roughening surface, and it correlates with the disappearance of the surface fractal character [37]. Talu et al. studied the influence of artificial saliva storage on 3-D surface texture characteristics of dental nanocomposites by using multifractal analysis. The exposure of artificial saliva storage changes anisotropic surface texture to more isotropic [55]. Glass ionomer surface roughness was investigated by Reddy et al. They used one-way ANOVA and Tukey’s significant difference tests to compare the surface of the material after conditioning it in citric acid of a different pH. The effects of pH on the surface texture of a glass ionomer cement depend on the used material and cause deterioration of a type II glass ionomer cement [56]. FDA is also used in the analysis of the surface of prosthetic restorations. Schestatsky et. al. demonstrated that the CAD/CAM technique produced smoother but more complex topography features (higher FD) than the pressing technique [35].

FDA is also used in dentistry to test organic tissues, such as teeth or bones. In a work by Nezefat et al., molar teeth with their enamel, the dentin, and cementum were tested using a power spectral density analysis and fractal dimension through AFM images. Hayek et al. tested radiographic estimation of bone quality using FDA. They noticed that it was a useful and non-invasive tool for examining bone density before the implantation, and proposed an image-based classification of bone density in the posterior regions of each jaw [57]. FDA can also be used as an objective method for detecting bone destruction induced by periodontitis. In the study conducted by Belgin et al., it was demonstrated that the FD values in patients with periodontitis were significantly lower than in the group without the disease [58].

Among the materials studied, Ketac Fil Plus (3M ESPE, Germany), Adhesor Carbofine (Pentron, Czech Republic), and Riva Self Cure (SDI, Australia) had the most extensive surface areas. This is indicated by a high value of TI and BI indexes. At the other end are the materials Agatos S (Chema-Elektromet, Poland) and Breeze (Pentron Clinical, USA). The latter cement, in particular, has a very smooth, homogeneous surface.

## 5. Conclusions

The Ketac Fil Plus (3M ESPE, Germany) and Riva Self Cure (SDI, Australia) cements provided the most favorable conditions for fibroblast adhesion, despite the statistically significant differences in the fractal dimension between them. Insofar as the surface texture is considered, Ketac Fil Plus (3M ESPE, Germany) cement should be indicated here.Ketac Fil Plus (3M ESPE, Germany) and Riva Self Cure (SDI, Australia) also showed no cytotoxicity potential. In contrast, Adhesor Carbofine (Pentron, Czech Republic) caused a severe cytotoxicity effect. Other cements, Agatos S (Chema-Elektromet, Poland), Harvard Polycarboxylat Cement (Harvard Dental, Great Britain), and Breeze (Pentron Clinical, USA), showed a moderate cytotoxicity effect on Balb/3T3 cells.Moderate positive correlation was observed between fractal dimension (FD) and the amounts of cells after 48 h. The correlation coefficient was decreased to a weak positive linear after 5 days.The study revealed a moderate negative linear correlation between texture index (TI), Bone index (BI), and the amounts of cells after 48 h of incubation. The correlation coefficient decreased to weak negative linear after 5 days.

## 6. Study Limitations

Flat surfaces of cements were investigated to simplify taking microscopic photos for analysis. In real restorations, the surface is most commonly a cylinder; this may affect fibroblast culturing.Some of the examined cements: Ketac Fil Plus (3M ESPE, Germany) and Riva Self Cure (SDI, Australia) may release fluoride ions which may affect cytotoxicity.

## Figures and Tables

**Figure 1 materials-14-05857-f001:**
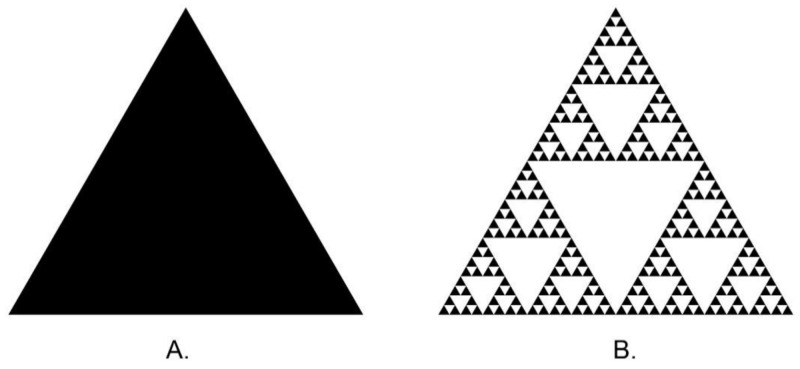
(**A**) Triangle (fractal dimension = 2), (**B**) Sierpinski triangle (fractal dimension ≈ 1.585) (Generated by https://codinglab.huostravelblog.com/math/fractal-generator/ Accessed on 1 October 2021).

**Figure 2 materials-14-05857-f002:**
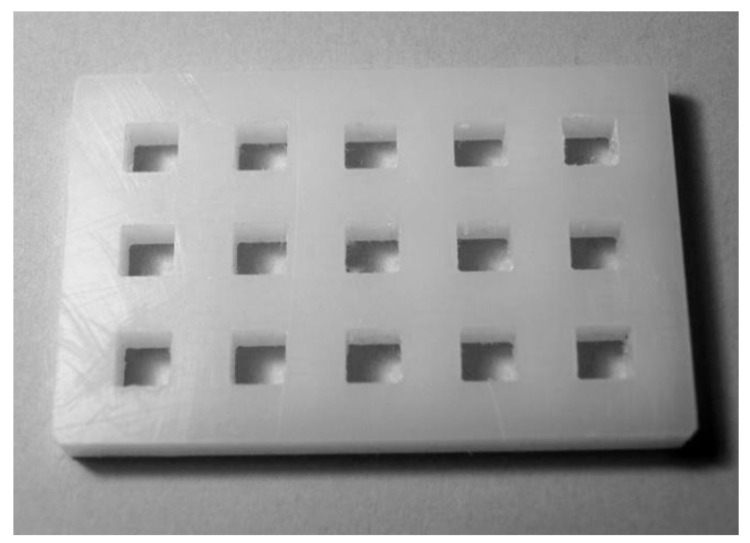
A plastic plate constituting the matrix for the production of test material cubes.

**Figure 3 materials-14-05857-f003:**
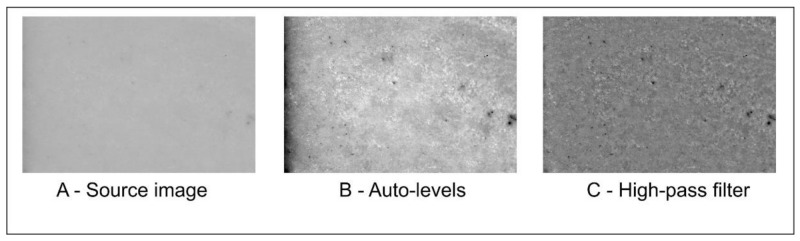
Graphical operations. (**A**) Source image, (**B**) image after auto-levels, (**C**) high-pass filter application.

**Figure 4 materials-14-05857-f004:**
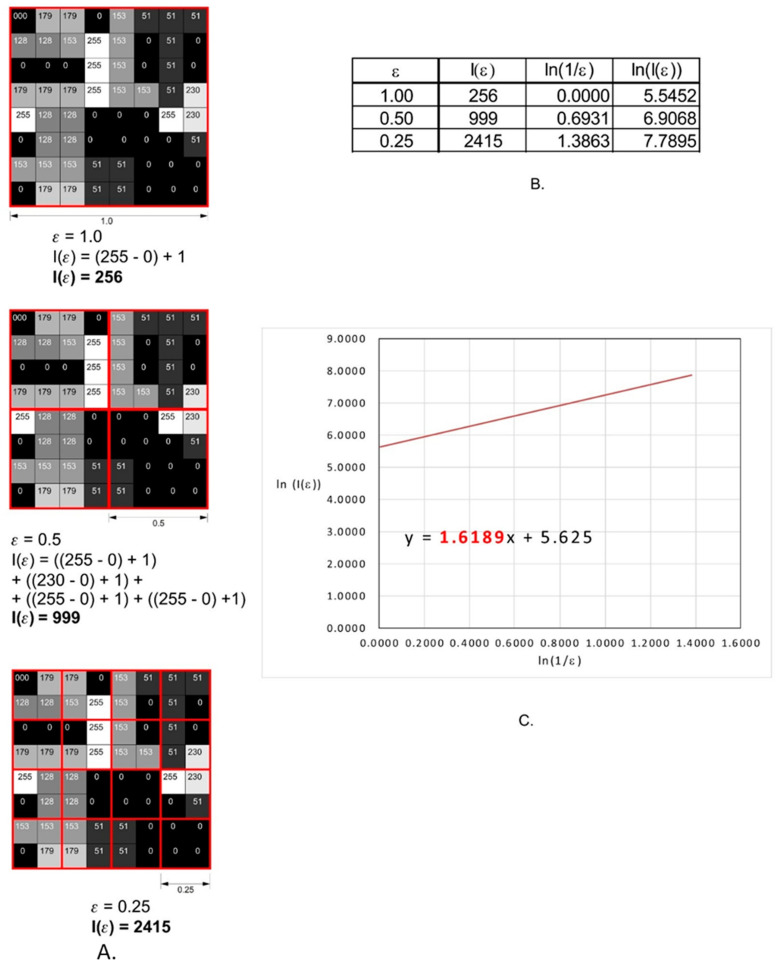
Graphical interpretation of the intensity difference algorithm of the fractal dimension calculation. (**A**) An example of a grayscale 8-bit image (8 × 8 pixels), the numbers in squares represent the intensity level of each pixel: 0, black, 255, white. The red squares represent the scale, ε. (**B**) The values of the intensity difference for each step of scale reduction (ε). (**C**) A straight line drawn through the points from table B on the x–y chart in a natural logarithm scale. The slope factor of this straight line is a value fractal dimension counted using intense difference algorithm.

**Figure 5 materials-14-05857-f005:**
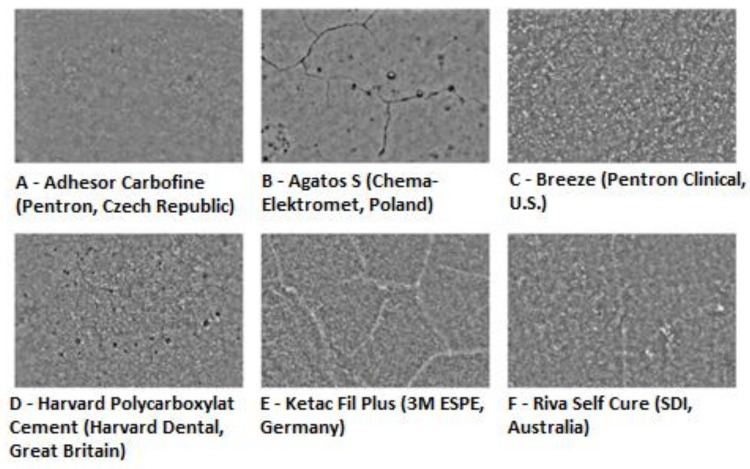
Examples of light microscope images of the surface of the tested cements subjected to subsequent analysis of textures and the fractal dimension (magnified 18 times).

**Figure 6 materials-14-05857-f006:**
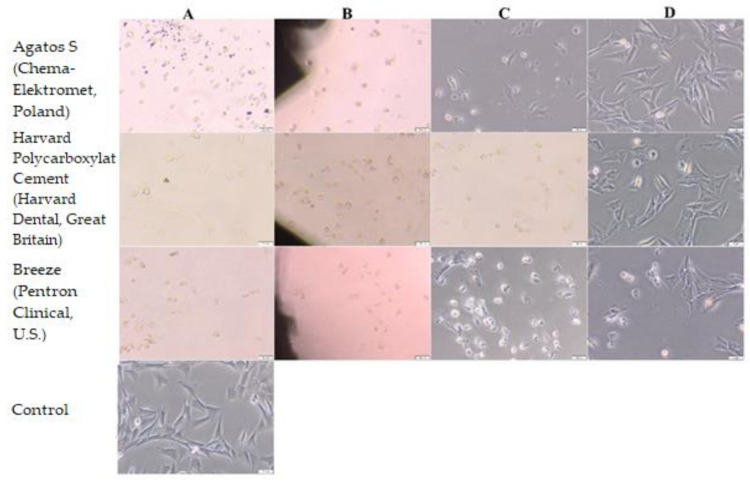
Morphology of Balb/3T3 cells after 24 h of contact with Agatos S (Chema-Elektromet, Poland), Harvard Polycarboxylat Cement (Harvard Dental, Great Britain) and Breeze (Pentron Clinical, USA); (**A**) under and (**B**) near the cement, (**C**) in area up to 1 cm, (**D**) in area more than 1 cm. Control culture had no contact with the test materials. Magn. 100×.

**Figure 7 materials-14-05857-f007:**
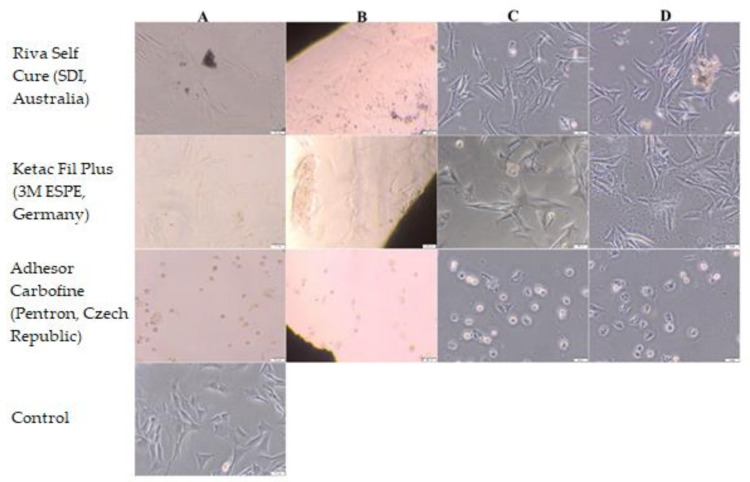
Morphology of Balb/3T3 cells after 24 h of contact with Riva Self Cure (SDI, Australia), Ketac Fil Plus (3M ESPE, Germany) and Adhesor Carbofine (Pentron, Czech Republic); (**A**) under and (**B**) near the cement, (**C**) in area up to 1 cm, (**D**) in area more than 1 cm. Control culture had no contact with the test materials. Magn. 100×.

**Figure 8 materials-14-05857-f008:**
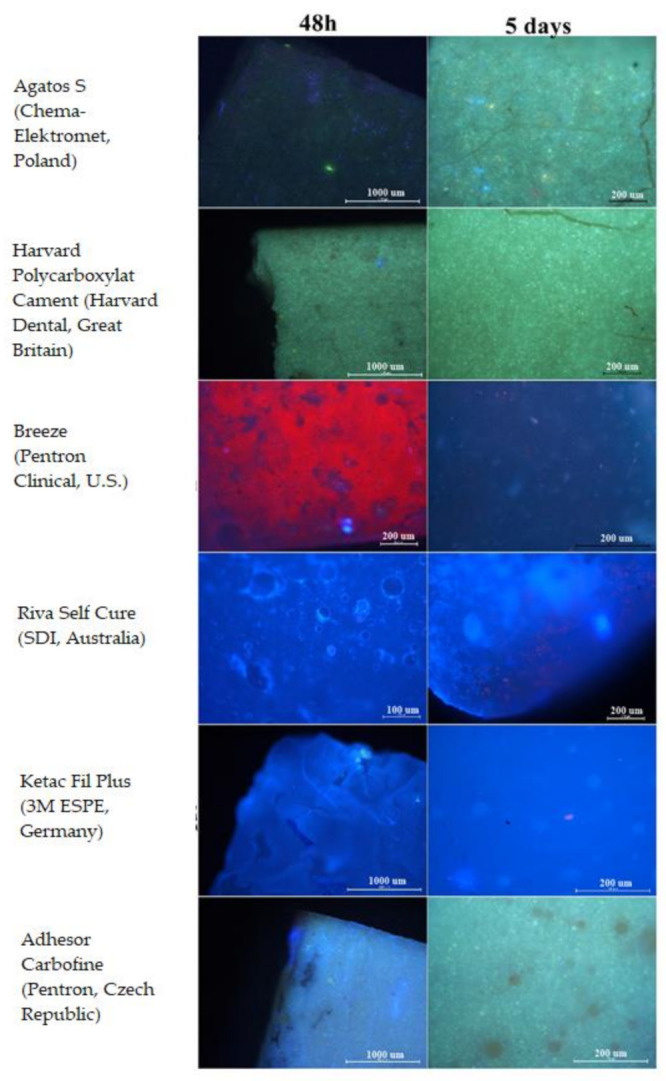
Adhesion of NHDF cells on cement surface after 48 h and 5 days.

**Figure 9 materials-14-05857-f009:**
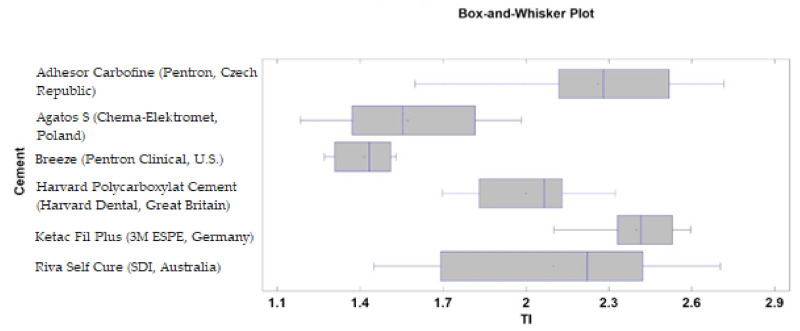
Texture index (TI) calculated for tested cements. The most expanded surface can be seen in Adhesor Carbofine (Pentron, Czech Republic) and Ketac Fil Plus (3M ESPE, Germany) contrary to Agatos S (Chema-Elektromet, Poland) and Breeze (Pentron Clinical, USA) presenting the lowest developed surface (*p* < 0.05).

**Table 1 materials-14-05857-t001:** Mean values of FD for individual cements and results of fibroblast adhesion tests after 48 h and 5 days of culture as well as evaluation of cytotoxicity of the tested cements (SD—standard deviation, FD—fractal dimension). Qualitative properties of cells were converted to quantitative parameters. For amounts of cells: lack of cells—0, few cells—1, cells on surface—2, culture of cells 3; for cytotoxicity: none—0, moderate—1, severe—2.

Cement	Adhesor Carbofine (Pentron, Czech Republic)	Agatos S (Chema-Elektromet, Poland)	Breeze (Pentron Clinical, USA)	Harvard Polycarboxylat Cement (Harvard Dental, Great Britain)	Ketac Fil Plus (3M ESPE, Germany)	Riva Self Cure (SDI, Australia)
**Mean FD**	1.593 ± 0.033	1.554 ± 0.019	1.425 ± 0.013	1.577 ± 0.032	1.590 ± 0.031	1.565 ± 0.038
**Amount of cells after 48 h**	lack of cells--- (0)	few cells--- (1)	lack of cells--- (0)	few cells--- (1)	cluster of cells (3)	few cells--- (1)
**Amount of cells after 5 days**	lack of cells--- (0)	few cells--- (1)	lack of cells--- (0)	lack of cells--- (0)	cells on surface (2)	cluster of cells (3)
**Cytotoxicity**	severe--- (2)	moderate--- (1)	moderate--- (1)	moderate--- (1)	none--- (0)	none--- (0)

**Table 2 materials-14-05857-t002:** Mean value and standard deviation fractal dimension (FD). Superscript indicates significant difference (post hoc ANOVA test, least significant difference, *p* < 0.05) to tested cement: ^1^ Adhesor Carbofine (Pentron, Czech Republic), ^2^ Agatos S (Chema-Elektromet, Poland), ^3^ Breeze (Pentron Clinical, USA), ^4^ Harvard Polycarboxylat Cement (Harvard Dental, Great Britain), ^5^ Ketac Fil Plus (3M ESPE, Germany), and ^6^ Riva Self Cure (SDI, Australia).

	Cement	FD
1	Adhesor Carbofine (Pentron, Czech Republic)	1.593 ± 0.030 ^2,3,6^
2	Agatos S (Chema-Elektromet, Poland)	1.554 ± 0.019 ^1,3,5^
3	Breeze (Pentron Clinical, USA)	1.425 ± 0.013 ^1,2,4,5,6^
4	Harvard Polycarboxylat Cement (Harvard Dental, Great Britain)	1.577 ± 0.032 ^3^
5	Ketac Fil Plus (3M ESPE, Germany)	1.590 ± 0.031 ^2,3,6^
6	Riva Self Cure (SDI, Australia)	1.565 ± 0.038 ^1,3,5^

**Table 3 materials-14-05857-t003:** Assessment of cytotoxic activity of the tested materials—in vitro tests on the Balb/3T3 line.

Cement	Morphological Changes in Cell Culture	Cell Culture Evaluation	Cytotoxicity
Agatos S (Chema-Elektromet, Poland)	cells degeneration and lysis observed under the sample and in zone up to 1 cm around the sample	3	moderate
Harvard Polycarboxylat Cement (Harvard Dental, Great Britain)	cells degeneration and lysis observed under the sample and in zone up to 1 cm around the sample	3	moderate
Breeze (Pentron Clinical, USA)	cells degeneration and lysis observed under the sample and in zone up to 1 cm around the sample	3	moderate
Riva Self Cure (SDI, Australia)	no morphological changes in cells under and around the sample	0	none
Ketac Fil Plus (3M ESPE, Germany)	no morphological changes in cells under and around the sample	0	none
Adhesor Carbofine (Pentron, Czech Republic)	cells degeneration and lysis observed under the sample and in zone around the sample and over the entire surface of the well	4	severe

**Table 4 materials-14-05857-t004:** Texture analysis. Mean value and standard deviation in the calculated texture indices. Superscript indicates significant difference (*p* < 0.05) to tested cement: ^1^ Adhesor Carbofine (Pentron, Czech Republic), ^2^ Agatos S (Chema-Elektromet, Poland), ^3^ Breeze (Pentron Clinical, USA), ^4^ Harvard Polycarboxylat Cement (Harvard Dental, Great Britain), ^5^ Ketac Fil Plus (3M ESPE, Germany), and ^6^ Riva Self Cure (SDI, Australia).

	Cement	TI	BI
1	Adhesor Carbofine (Pentron, Czech Republic)	2.26 ± 0.37 ^2,3,4^	0.97 ± 0.18 ^2,3,4,6^
2	Agatos S (Chema-Elektromet, Poland)	1.57 ± 0.24 ^1,4,5,6^	0.64 ± 0.10 ^1,4,5,6^
3	Breeze (Pentron Clinical, USA)	1.41 ± 0.10 ^1,4,5,6^	0.56 ± 0.04 ^1,4,5,6^
4	Harvard Polycarboxylat Cement (Harvard Dental, Great Britain)	2.00 ± 0.19 ^1,2,3,5^	0.83 ± 0.09 ^1,2,3,5^
5	Ketac Fil Plus (3M ESPE, Germany)	2.40 ± 0.15 ^2,3,4,6^	0.99 ± 0.09 ^2,3,4,6^
6	Riva Self Cure (SDI, Australia)	2.10 ± 0.41 ^2,3,5^	0.87 ± 0.19 ^1,2,3,5^

## Data Availability

Data are available from the authors: katarzyna.skoskiewicz-malinowska@umed.wroc.pl, kamil.jurczyszyn@umed.wroc.pl.

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
