# Peer review of "Application of Texture and Fractal Dimension Analysis to Evaluate Subgingival Cement Surfaces in Terms of Biocompatibility"

_materials, 2021, doi:10.3390/ma14195857_

Round 1

Reviewer 1 Report

The investigation on the correlationship between the biocompatibility and surface morphology of a biomaterial is of significance for the developments of both surface science and medical instruments. In this study, the texture analysis and fractal dimension were both used to characterize the surface images of subgingival cement samples with 6 brands. However, the results and discussion of this paper are so inadequate that a recommendation for publication cannot be supported. A substantial revision is required in prior to a reconsideration of the reviewer.

First, the introduction section should be reorganized, because the logic for the necessity of this study is not clear. Moreover, the majority of the discussion section also introduced lots of details of the previous studies in literature, which should be moved to the introduction section.

Second, fractal geometrical was NOT introduced in 1983. Please check the publications by Mandelbrot.

Third, Fig. 1 is not suitable to appear in the paper, because it is not the creation by authors but could be found in textbooks.

Fourth, the variables in text and equations in subsection 2.1 are with poor formats. Please revise them according to the other sections.

Fifth, the subsections of section 2 should be in a reasonable sequence, e.g. preparing the cement samples--> Taking images-->Texture/ fractal analysis

Sixth, the third term of conclusion section is important because it is close to the paper title. However, its content (mediocre linear correlation between the mean value of the fractal dimension and texture indexes and the properties of the surface for fibroblast adhesion) is not clear and can not be demonstrated in the results, particularly the figures or tables.

Seventh, also in the third term of conclusion, what is the meaning and indications of “correlation coefficient decreased with time”? Please enhance the details in its corresponding results and discussions.

Eighth, both texture analysis and fractal dimension are relatively new in the morphology characterization. The calculations with traditional methods (such as surface roughness) should be also carried out and compared, in order to reveal the advantage of texture analysis and fractal dimension, which are the core progresses of this study.

Ninth, the English writing of this paper is rough, and much more work to polish the grammar errors/ inappropriate is required before the resubmission.

Author Response

Dear Reviewer,

According to Your valuable suggestion, we applied following changes:

First, the introduction section should be reorganized, because the logic for the necessity of this study is not clear. Moreover, the majority of the discussion section also introduced lots of details of the previous studies in literature, which should be moved to the introduction section.

We have made several changes to the manuscript, taking into consideration all the comments and suggestions that were made.

Second, fractal geometrical was NOT introduced in 1983. Please check the publications by Mandelbrot.

Of course, thank You for the comment. It was corrected in the manuscript.

Third, Fig. 1 is not suitable to appear in the paper, because it is not the creation by authors but could be found in textbooks.

We replaced this figure. The new one was generated on the web side: https://codinglab.huostravelblog.com/math/fractal-generator/

Fourth, the variables in text and equations in subsection 2.1 are with poor formats. Please revise them according to the other sections.

Corrected

Fifth, the subsections of section 2 should be in a reasonable sequence, e.g. preparing the cement samples--> Taking images-->Texture/ fractal analysis

It was corrected in the manuscript.

Sixth, the third term of conclusion section is important because it is close to the paper title. However, its content (mediocre linear correlation between the mean value of the fractal dimension and texture indexes and the properties of the surface for fibroblast adhesion) is not clear and can not be demonstrated in the results, particularly the figures or tables.

Pearson correlation coefficient “r” was calculated between FD, TI, BI and amounts of cells after 48 hours and 5 days. It is described in lines 248-252 and lines 343-355. Second Reviewer suggested to reduce number of tables so we add some information in the conclusion section to be more clarify.

Seventh, also in the third term of conclusion, what is the meaning and indications of “correlation coefficient decreased with time”? Please enhance the details in its corresponding results and discussions.

Amounts of cells were measured after 48 hours and 5 days. Correlation coefficient were counted up two times between FD and amounts of cells (after 48 hours and 5 days). The value of this coefficient was lower after 5 days than after 48 hours. So conclusion that correlation coefficient decreased with time. We added this to the conclusion section.

Eighth, both texture analysis and fractal dimension are relatively new in the morphology characterization. The calculations with traditional methods (such as surface roughness) should be also carried out and compared, in order to reveal the advantage of texture analysis and fractal dimension, which are the core progresses of this study.

The main aim of this study was an evaluation of biocompatibility cements in aspect of subgingival restorations. Measuring of amounts of fibroblasts and cytotoxicity was a core topic of this study. Some of studies use traditional methods of surface analysis. We wanted to use novel methods to evaluate a correlation between fractal dimension, texture properties and colonization of fibroblasts on various cement surfaces. In the future study we will compare our mathematical methods versus traditional.

Ninth, the English writing of this paper is rough, and much more work to polish the grammar errors/ inappropriate is required before the resubmission.

The manuscript was thoroughly revised to identify and eliminate most of grammar, spelling and stylistic issues.

Best regards,

Authors.

Reviewer 2 Report

My comments and suggestions are as follows:

Introduction: In my opinion, because one of the materials studied is Breeze, a luting resin-based material, you should mention this kind of resins in lines 65-70, not only restorative ones.

The aim of the study should be defined in the end of the introduction.

Line 56-63. When referring to GIC's, you should mention the luting type, too, as they come in direct contact with the gingiva, as well, not only the restorative type. 

Line 76: "zinc phosphate is used for metal-ceramic restorations, long-term fixed partial dentures, and cast post cores". You should mention for luting, as it is one could understand that the mentioned prosthetic restorations are made of zinc phosphate. Metal-ceramic restorations are included in the category of fixed partial dentures. The term "long-term fixed partial dentures" is not appropriate, please use fixed partial dentures or crowns and bridges. Specify only if they are provisional. 

Materials and methods:

2.5. Procedure for preparing the cement samples should be 2.1., as the materials should be described first, and the experiments following.

Please give manufacturer and location for each material, in brackets, each time you mention it.

Which kind of Ketac did you use? There are different types, with different purposes, for restorations, luting etc. Ketac Cem is a luting GIC, Ketac Molar is a restorative GIC, Ketac Cem Plus is a resin-modified GIC.

Results: Figure 6 and 7. Please define control. I could not find any information in the main text.

Line 364. Please define FDA when first used. At this point only FD was previously defined.

Lines 265-368 have no relevance in the context of your study. The information should be removed.

Line 370-373. Same as above. Shorten the phrase and keep only basic info. 

Line 379-380. Remove.

Line 381: "Among the materials studied, cements had the most extensive surface area...". All the materials studied are cements, of different types, not only Ketac, Adhesor and Riva. Please correct.

Line 384-387. "The result is in line with the systematic review carried out by Boing 384 et. al. They found that Resin-based composites have a better surface texture than Glass Ionomer Cement at all follow-up periods and explain it by lower abrasion resistance of GIC in comparison to Resin-based composites [18]." This is not quite true, as the above mentioned study deals with RESTORATIVE composites and GICs. As I mentioned before Breeze is a self-adhesive resin cement (for luting).

Which kind of Riva did you use: Riva Light-cure, Riva luting, Riva cem, Riva Luting Plus, Riva silver? The are different kinds, with different uses and properties, as in Ketac... Some are classic GIC's, some are resin modified or reinforced. 

Agatos S was mentioned in the abstract and once in the main text, than the term Agatos was used. There is another type of Agatos, W, please use the correct name each time.

There are several Harvard products (cements): zync phosphate, polycarboxilate, GIC, temporary. Please specify the type used.

In my opinion, the materials should be mentioned properly, in the current form, the article is misleading and is not suitable for publication. These information are basic and have to be clarified. 

Author Response

Dear Reviewer,

According to Your valuable suggestions we applied following changes:

Introduction: In my opinion, because one of the materials studied is Breeze, a luting resin-based material, you should mention this kind of resins in lines 65-70, not only restorative ones.

Added: Resin-based materials are used also as luting cements, often in subgingival preparations with challenging field isolation. The greatest advantages of resin luting materials are: improved mechanical properties, lower solubility and reinforcement of all-ceramic restorations in comparison with the traditional luting cements.

The aim of the study should be defined in the end of the introduction.

Added: Application of Texture and Fractal Dimension Analysis to evaluate subgingival cement surfaces in terms of biocompatibility

Line 56-63. When referring to GIC's, you should mention the luting type, too, as they come in direct contact with the gingiva, as well, not only the restorative type. 

Added: Cements based on glass-ionomer materials are extremely popular definitive luting agent. The chemistry of the setting reaction is complex and take several months to reach completion.

Line 76: "zinc phosphate is used for metal-ceramic restorations, long-term fixed partial dentures, and cast post cores". You should mention for luting, as it is one could understand that the mentioned prosthetic restorations are made of zinc phosphate. Metal-ceramic restorations are included in the category of fixed partial dentures. The term "long-term fixed partial dentures" is not appropriate, please use fixed partial dentures or crowns and bridges. Specify only if they are provisional. 

We have made several changes to the manuscript, taking into consideration all the comments and suggestions that were made:

Due to its early strength, low cost, acceptable physical properties, and easy clinical use, zinc phosphate is used for luting fixed partial dentures including metal-ceramic resto-rations and cast post cores [22].

Materials and methods:

2.5. Procedure for preparing the cement samples should be 2.1., as the materials should be described first, and the experiments following.

We have made several changes to the manuscript, taking into consideration all the comments and suggestions that were made.

Please give manufacturer and location for each material, in brackets, each time you mention it.

We have made several changes to the manuscript, taking into consideration all the comments and suggestions that were made.

Which kind of Ketac did you use? There are different types, with different purposes, for restorations, luting etc. Ketac Cem is a luting GIC, Ketac Molar is a restorative GIC, Ketac Cem Plus is a resin-modified GIC.

Thank You very much for this suggestion. The type of material has been specified in detail - Ketac™ Fil Plus (3M ESPE, Germany).

Results: Figure 6 and 7. Please define control. I could not find any information in the main text.

The control was incubation of cell culture without any cements. This information was added to the figure 6 and 7 caption.

Line 364. Please define FDA when first used. At this point only FD was previously defined.

FDA is defined in 86 line.

Lines 265-368 have no relevance in the context of your study. The information should be removed.

In these lines results of biological evaluation of cements and texture analysis are describe. Cytotoxicity is a part of our study. Without estimation of cytotoxicity and amounts of cells we wouldn’t be able to corelate these data with fractal dimension. The same texture analysis is a base of our study.

It is also worth noting that cell adhesion depends on the state of surface development [this is what dental plaque adhesion is based on]. It seems that the determination of the surface structure characteristics of the tested materials, which will be clinically exposed to the oral flora, is important and needed in the evaluation of cements.

Some of less important sentences were removed.

Line 370-373. Same as above. Shorten the phrase and keep only basic info. 

It was shorten.

Line 379-380. Remove.

It was removed.

Line 381: "Among the materials studied, cements had the most extensive surface area...". All the materials studied are cements, of different types, not only Ketac, Adhesor and Riva. Please correct.

Of course, thank You for the comment. It was corrected in the manuscript.

Line 384-387. "The result is in line with the systematic review carried out by Boing 384 et. al. They found that Resin-based composites have a better surface texture than Glass Ionomer Cement at all follow-up periods and explain it by lower abrasion resistance of GIC in comparison to Resin-based composites [18]." This is not quite true, as the above mentioned study deals with RESTORATIVE composites and GICs. As I mentioned before Breeze is a self-adhesive resin cement (for luting).

Thank You for this remark. We have decided to remove this part of text.

Which kind of Riva did you use: Riva Light-cure, Riva luting, Riva cem, Riva Luting Plus, Riva silver? The are different kinds, with different uses and properties, as in Ketac... Some are classic GIC's, some are resin modified or reinforced. 

Thank You very much for this suggestion. The type of material has been specified in detail - Ketac™ Fil Plus (3M ESPE, Germany), Riva Self Cure (SDI, Australia).

Agatos S was mentioned in the abstract and once in the main text, than the term Agatos was used. There is another type of Agatos, W, please use the correct name each time.

Thank You very much for this suggestion. The type of material has been specified in detail – Agatos S (Chema-Elektromet, Poland).

There are several Harvard products (cements): zync phosphate, polycarboxilate, GIC, temporary. Please specify the type used.

Thank You very much for this suggestion. The type of material has been specified in detail – Harvard Polycarboxylat Cement (Harvard Dental, Great Britain).

In my opinion, the materials should be mentioned properly, in the current form, the article is misleading and is not suitable for publication. These information are basic and have to be clarified.

Reviewer 3 Report

Results. (1) In Table 1, do the values of fractal dimensions have units, such as micrometers or nanometers? This point should be noted in the text, with a brief explanation. (2) In Tables 1 and 2, are too many significant figures being used to report the fractal dimensions (means and standard deviations) and the ANOVA results, respectively. Brief comments in the text about the appropriate number of significant figures would be worthwhile. Care and consistency are important when deciding about the number of significant figures being presented for the data in the text as well.

Discussion. (1) Is it possible to make comments about the differing microstructural features contained within the individual fractals for the different cements? (Perhaps another study would be needed to obtain this detailed information.) (2) What are the limitations of the study? What should be investigated in future research?

Author Response

Dear Reviewer,

According to Your valuable suggestions we applied following changes:

 (1) In Table 1, do the values of fractal dimensions have units, such as micrometers or nanometers? This point should be noted in the text, with a brief explanation.

Fractal dimension has no unit. For example dimension of line equals 1 (it is number of dimensions not a value of length). In Euclidian geometry we used to know that amounts of dimensions are an integer. Fractal dimension of 2 dimensioned forms are the fractions between 2 and 3.  

This information has been added to the introduction.

(2) In Tables 1 and 2, are too many significant figures being used to report the fractal dimensions (means and standard deviations) and the ANOVA results, respectively. Brief comments in the text about the appropriate number of significant figures would be worthwhile. Care and consistency are important when deciding about the number of significant figures being presented for the data in the text as well.

In table 1 we tried to show as most as possible results to not multiple tables. We remove one row which shows SD.

We rebuilt table 2, to be more clarify.

Discussion.

  • Is it possible to make comments about the differing microstructural features contained within the individual fractals for the different cements? (Perhaps another study would be needed to obtain this detailed information.)

Yes, we agree that next study is needed to obtain more information about fractal dimension analysis in aspect of individual cements.

  • What are the limitations of the study? What should be investigated in future research?

Limitations:

  1. Flat surfaces of cements were investigated to simplify taking microscopic photos for analysis. In real restorations’ surface is most common section of cylinder, this may affect for fibroblast culturing.
  2. Some of examined cements may release fluoride ions which may affect for cytotoxicity. (We added these informations to the manuscript).

In the future, this study will be the basis for evaluating changes in the surface appearance [texture analysis] of materials after fluoride hydration.

Manuscript was checked by professional translator, we attached translate certificate as separate file.

Best regards,

Authors.

Round 2

Reviewer 1 Report

(1) There are still format issues in subsection 2.1.

(2) Figure 2 is quite unfriendly for readers. All the subfigures are not clear, and many variables are not italic. Moreover, is it necessary to include the equations and table in this figure?

(3) The demonstration for the third term of conclusion section is still confusion. The details mentioned in reply “described in lines 248-252 and lines 343-355” could not be found in the manuscript.

(4) In subsection 3.1, the results require a reorganization. The numbers of Tables 1 and 2 are the same, verifying that “Second Reviewer suggested to reduce number of tables”.

(5) “24 hours” is stated in line 306, while “48 hours” is mentioned in other locations.

(6) The first paragraph in subsection 3.1 indicated that the correlation coefficient between the fractal dimension and the number of fibro-blasts could be quantified as r=0.47 and 0.31 according to the results in Table 1. However, merely “lack of cells” and “few cells” were in Table 1 instead of quantified values. Therefore, the detailed results might be necessary for this subsection, and one more figure about the calculation of correlation coefficient should be added to enhance the demonstration.

Author Response

Dear Reviewer,

 W applied following changes according to your valuable suggestions:

  • There are still format issues in subsection 2.1.

It was corrected in the manuscript. An order was changed to following sequence:  Procedure for preparing the cement samples, Biological evaluation, Taking images, Fractal dimension analysis, Texture analysis, Statistical analysis.

  • Figure 2 is quite unfriendly for readers. All the subfigures are not clear, and many variables are not italic. Moreover, is it necessary to include the equations and table in this figure?

Figure 2 (in version of manuscript which was attached in round 1) shows a plastic plate constituting the matrix for the production of test material cubes. I try to guess You are in mind figure 4. In our opinion this figure with fully description in manuscript is clear as possible for readers. In our previously study (accepted to publish) this figure was divided into table and chart separately but on the stage of editorial corrections we were asked do consolidate these parts into one figure. So in this study we prepared this image with small modification (to avoid self citation) like in previously publication to avoid editorial corrections.  We added information that it is an example of analysis 8x8 pixels image.

  • The demonstration for the third term of conclusion section is still confusion. The details mentioned in reply “described in lines 248-252 and lines 343-355” could not be found in the manuscript.

Detailed description was added M&M section (statistical analysis). 

(4) In subsection 3.1, the results require a reorganization. The numbers of Tables 1 and 2 are the same, verifying that “Second Reviewer suggested to reduce number of tables”.

Table 1 shows FD (mean and SD) in aspect of cell properties (amounts of cells and cytotoxicity). Table 2 shows results of statistical tests between FD values (modified according to suggestion of second reviewer. In this format it is complementary to format of table 4.

(5) “24 hours” is stated in line 306, while “48 hours” is mentioned in other locations.

It was corrected in the manuscript. Number of cells were examined after 48 hours and 5 days. Cytotoxicity was checked after 24 hours.

(6) The first paragraph in subsection 3.1 indicated that the correlation coefficient between the fractal dimension and the number of fibro-blasts could be quantified as r=0.47 and 0.31 according to the results in Table 1. However, merely “lack of cells” and “few cells” were in Table 1 instead of quantified values. Therefore, the detailed results might be necessary for this subsection, and one more figure about the calculation of correlation coefficient should be added to enhance the demonstration.

We agree it was not clearly stated. Qualitive properties of cells was converted to quantitative parameters. For amounts of cells:  lack of cells – 0, few cells -1, cells on surface – 2, culture of cells 3; for cytotoxicity: none – 0, moderate -1, severe -2. These parameters were added to table 1. Correlation coefficient was calculated between FD value and these quantitative values. This information was added to the caption of table and in M&M section.

Best regards,

Authors.

Reviewer 2 Report

I have no further comments. 

Thank you for improving your manuscript.

Author Response

Thank You,

Best regards,

Authors.